# Non-Chemical Treatments for the Pre- and Post-Harvest Elicitation of Defense Mechanisms in the Fungi–Avocado Pathosystem

**DOI:** 10.3390/molecules26226819

**Published:** 2021-11-11

**Authors:** Juan Antonio Herrera-González, Silvia Bautista-Baños, Mario Serrano, Gianfranco Romanazzi, Porfirio Gutiérrez-Martínez

**Affiliations:** 1Laboratorio Integral de Investigación en Alimentos, TecNM-Instituto Tecnológico de Tepic, Av. Tecnológico 2595, Lagos de Country, Tepic 63175, Mexico; juanherrerago@ittepic.edu.mx; 2Instituto Nacional de Investigaciones Forestales, Agrícolas y Pecuarias, Campo Experimental Uruapan, Av. Latinoamericana 1101, Col. Revolución, Uruapan 60150, Mexico; 3Centro de Desarrollo de Productos Bióticos, Instituto Politécnico Nacional, Carretera Yautepec-Jojutla Km 6, CEPROBI 8, San Isidro, Yautepec 62730, Mexico; sbautis@ipn.mx; 4Centro de Ciencias Genómicas, Universidad Nacional Autónoma de Mexico, Cuernavaca 62209, Mexico; serrano@ccg.unam.mx; 5Department of Agricultural, Food and Environmental Sciences, Marche Polytechnic University, 60131 Ancona, Italy; g.romanazzi@univpm.it

**Keywords:** *Bacillus*, postharvest, chitosan, essential oils, silicon, *Colletotrichum* spp.

## Abstract

The greatest challenge for the avocado (*Persea americana* Miller) industry is to maintain the quality of the fruit to meet consumer requirements. Anthracnose is considered the most important disease in this industry, and it is caused by different species of the genus *Colletotrichum*, although other pathogens can be equally important. The defense mechanisms that fruit naturally uses can be triggered in response to the attack of pathogenic microorganisms and also by the application of exogenous elicitors in the form of GRAS compounds. The elicitors are recognized by receptors called PRRs, which are proteins located on the avocado fruit cell surface that have high affinity and specificity for PAMPs, MAMPs, and DAMPs. The activation of defense-signaling pathways depends on ethylene, salicylic, and jasmonic acids, and it occurs hours or days after PTI activation. These defense mechanisms aim to drive the pathogen to death. The application of essential oils, antagonists, volatile compounds, chitosan and silicon has been documented in vitro and on avocado fruit, showing some of them to have elicitor and fungicidal effects that are reflected in the postharvest quality of the fruit and a lower incidence of diseases. The main focus of these studies has been on anthracnose diseases. This review presents the most relevant advances in the use of natural compounds with antifungal and elicitor effects in plant tissues.

## 1. Introduction

The greatest challenge for the avocado (*Persea americana* Miller) industry is to maintain the quality of the fruit in order to meet consumer requirements, e.g., in terms of shape, skin color, and texture, among others. In addition, the degree of perishability of the product is another attribute that has to be considered. However, one of the main constraints when it comes to achieving these characteristics is the incidence of fungal diseases during fruit storage. Pre- and post-harvest losses due to phytopathogenic fungi diseases can be as high as 80% [1,2,3]. To date, anthracnose, caused by different species of the genus *Colletotrichum*, is considered the most important disease in this industry [4,5,6]. To control these fungi, the main method has been the use of commercial fungicides, such as azoxystrobin and azoxystrobin + fludioxonil. However, their application has been restricted or banned due to their toxicity to humans and to the environment as well as fungal resistance generation.

Intrinsically, most agricultural products possess defense mechanisms that can be triggered in response to the attack of pathogenic microorganisms and can be induced by the application of exogenous compounds that are Generally Recognized as Safe (GRAS). The defense mechanisms have an elicitor effect, which includes the activation of induced local resistance (only in the infected area) and induced systemic resistance (in the whole fruit) (Figure 1) [7,8,9]. Information on the defense mechanisms of the fruit and those compounds with elicitor activity on the avocado fruit is scarce and dispersed, focusing mainly on physiological and quality aspects of the treated fruit. Unfortunately, they tend to omit the enzymatic and molecular activation processes. The objective of this review is to gather published information about the natural defense mechanisms and elicitation occurrence on avocado fruit by different natural substances considered as GRAS compounds, as well as their antifungal effect (modes action and target sites) on *Colletotrichum gloeosporioides*.

## 2. Mechanisms of Resistance

To understand the mode of action and target sites of alternative methods to synthetic fungicides, it is necessary to know the defense mechanisms that fruits or plants naturally activate against the attack of a phytopathogen [10]. During the infection process, the pathogen firstly has to penetrate the fruit cuticle. According to Tafolla-Arellano et al. [11] and Camacho-Vázquez et al. [12], this is composed of two layers of epicuticular waxes (esters, alcohols, aldehydes, ketones, and long-chain fatty acids) of amorphous and crystalline forms, and cutin and intracuticular waxes. When the phytopathogen achieves this penetration, in order to maintain contact with the plasma membrane, the cell wall must use inducible defense mechanisms that provide a specific resistance through the activation of the innate immune system of the plant called PTI (PAMP-Triggered Immunity) [12,13,14]. PTI starts after the appearance of elicitors commonly known as PAMPs (Pathogen-Associated Molecular Patterns), DAMPs (Damage-Associated Molecular Patterns), and MAMPs (Microbe-Associated Molecular Patterns) [13,15,16]. In general, these elicitors are recognized by receptors called PRRs (Pattern Recognition Receptors), which are proteins located on the cell surface that have high affinity and specificity for PAMPs, MAMPs, and DAMPs in concentrations below nanomoles [17,18].

After this recognition—a process that takes seconds or minutes—a rapid diffusion of ions is generated through the plasma membrane, and the concentration of intracellular Ca^2+^ increases, while MAPKs (Mitogen-Activated Protein Kinases) and CDPKs (Calcium-Dependent Protein Kinases) are activated, as is the production of ROS (Reactive Oxygen Species). The production of ROS, which can exist independently with one, two, or three unpaired electrons and with an average life of 10 s, is activated 1 or 2 h after infection. The most common free radical compounds of ROS are H_2_O_2_ (hydrogen peroxide), ^1^O_2_ (single oxygen), and OH• (hydroxyl radicals). These compounds can oxidize host and pathogen cellular components and lead to oxidative destruction through peroxidation, protein oxidation, inhibition of enzyme activities, and DNA or RNA damage [19,20,21,22].

The activation of defense-signaling pathways dependent on ethylene, salicylic, and jasmonic acids occurs hours or days after PTI activation [23,24]. This activation results in the induction of late response genes, such as PR (Pathogen-Related) genes, as well as the accumulation of toxic secondary metabolites and the production of histological barriers, such as callose or lignin. Lignification is a process of binding cell wall proteins and wound healing, which provides mechanical resistance, rigidity, and hydrophobicity to the secondary cell walls for the transport of water and nutrients, as well as protease inhibitors. For example, this response is documented in the early stages of the infection by *C. gloeosporioides* [25] and is part of a complex process that involves phenolic compounds and peroxides that heal and prevent future attacks by other plant pathogens [26,27]. Subsequently, the production of phytoalexins occurs. These are synthesized in the periphery of infected or dying cells and accumulate in sufficient concentrations to inhibit the fungus and kill plant cells close to the infection. The phytoalexin biosynthesis is transient, reaching its highest concentration a few hours after infection. Additionally, a pH change occurs (an increase in H^+^ and Ca^2+^ in the cytosol, acidification of the cytoplasm, depolarization of the membrane, and protein phosphorylation) [28], and, finally, the enzymes peroxidase, phenylalanine ammonia-lyase (PAL), chitinase, and β-1, 3-glucanase, which are responsible for the synthesis of the most important polysaccharides of the cell wall, are activated [29]. These defense mechanisms aim to kill the pathogen [30,31]. However, in fruit, as the process of ripening to maturity of consumption progresses, these processes become inactive.

## 3. Alternative Control Methods That Involve the Activation of Defense Mechanisms in Avocado

Alternative methods to the use of synthetic fungicides are compounds that usually have an antifungal and/or elicitor effect (GRAS compounds), are recognized by plant cells, and trigger the defense mechanisms of the fruit, including hypersensitive response. Next, some methods tested on avocado, and the pathogens isolated from this fruit, are considered.

### 3.1. Essential Oils

Essential oils are on the list of GRAS additives for human consumption. The antifungal and elicitor activity of essential oils—complex substances of secondary metabolites—has been widely studied. Due to their hydrophobic nature, they can penetrate the phospholipids of the *C. gloeosporioides* cell wall, altering their permeability. This leads to an outflow of ions (calcium, Ca^2+^) and intracellular liquid (radicals, cytochrome C, and proteins). Once inside the *C. gloeosporioides*, essential oils alter the flow of electrons, the motive force of protons, and the active transport and coagulation of cellular content. They reduce the pH, alter the processes of respiration and energy production, the synthesis of cellular components, and the loss of cell homeostasis. In addition, the fungus changes the permeability of the mitochondrial membrane, and, finally, the pathogen dies [32,33,34]. In terms of these effects, essential oils are considered bio-fungicides with multiple target sites and modes of action. The elicitor action of essential oils is based on the increase in the expression of phenylalanine ammonium lyase genes. The synthesis of this enzyme is the most important factor in the phenylpropanoid pathway. It synthesizes phenolic compounds (phenolic acid and flavonoids) that improve their antioxidant capacity, making the “Hass” avocado fruit more resistant to attack by *C. gloeosporioides* [35]. The application of thyme, peppermint, and citronella essential oils in low concentrations (10–500 µL/L) did not alter the sensory or generated microbial resistance in “Hass” and “Fuerte” avocado fruit previously inoculated with *C. gloeosporioides* [33,36,37,38,39,40] (Table 1 and Table 2).

### 3.2. Bacillus *spp.*

The *Bacillus* species have been used as biocontrol agents due to their action mechanisms as biofungicides (competition, parasitism, predators, and antagonism) and adaptation to a large number of environments [41,42]. They produce a large number of secondary metabolites with antagonistic activity, and they secrete antifungal proteins (antimicrobial, low toxicity, strong antimicrobial activities, high biodegradability, and high-temperature tolerance) and low molecular weight volatile compounds with antifungal activity [43]. In plant cells, the volatile compounds produced by the *Bacillus* do not have a toxic effect. Rather, they have an elicitor effect because they are perceived as signals for the activation of defense mechanisms. In preharvest treatments, they promote plant growth, secrete antimicrobial compounds and growth hormones, solubilize mineral phosphate, and chelate toxic metals [44,45]. For example, the application of *B. subtilis* in preharvest has been shown to colonize large areas of avocado trees and prevent the colonization of complex of fungal pathogens causing anthracnose and stem-end rot (*Colletotrichum* species, *Lasiodiplodia theobromae*, *Phomopsis perseae*, and *Dothiorella aromatica*) in avocado fruit. *B. subtilis* also survives in sufficiently large populations to control these postharvest diseases through mycoparasitism and competitive colonization [46]. The direct application of *Bacillus* on the avocado fruit has been recognized as GRAS. The metabolites produced by *Bacillus* make them good biocontrol agents that can replace synthetic fungicides. For example, preharvest applications of *B. subtilis* B246 on avocado flowers means that they can adhere, colonize, and survive effectively in the fruit. In addition, they adhere to the conidia and hyphae of the fungi *D. aromatica*, *C. gloeosporioides*, and *P. perseae* and cause lysis in the hyphae, degradation of conidia (parasitism), and inhibition of the germination of conidia by exclusion and preventive colonization [47,48] (Table 1 and Table 2).

### 3.3. Volatile Organic Compounds

Volatile compounds are organic compounds or solvents with lipophilic activity (i.e., the ability to dissolve fatty acid and lipids) and volatile properties at room temperature. They are classified based on their functional group (aliphatic, aromatic, alcohols, aldehydes, esters, among others) [49,50]. Volatile compounds extracted from plants and microorganisms have gained increasing global interest due to their volatility, safety, environmental friendliness, and antifungal properties. In addition, they have been classified as a GRAS substance used as an additive [51,52]. The antifungal action of volatile compounds is based on their hydrophobicity, which allows them to penetrate the cell membrane (H^+^ and K^+^ cations). Once inside, they dissolve the lipid phase of the cytoplasm. The membrane loses permeability due to the loss of the pH gradient and electrical potential in pathogens such as *L. theobromae* and *C. gloeosporioides*. This is followed by intracellular imbalance, osmotic pressure, organelle degradation, leakage of intracellular fluid, and loss of membrane permeability, which cause the death of these pathogens. In addition, these compounds can form hydrogen bonds with intra- and extra-cellular enzymes, interfering with enzymes that generate energy (ATP) and their substrate [36,53]. The mode of action of volatile compounds can be summarized in terms of the disorganization of the activities of the cell membrane, the decrease in the entry and exit of compounds (sodium and potassium pump), and the inhibition of the entry of oxygen, which alters oxidative phosphorylation (ATP production). Volatile compounds in plant cells have an elicitor effect. They are perceived as indicators of biotic or abiotic stress (pathogens, predator attack, or mechanical damage), which triggers the defense mechanisms of plant cells [52,54] (Table 1 and Table 2).

### 3.4. Chitosan (Coating, Elicitor, and Biofungicide)

Chitosan is a biopolymer derived from chitin (deacetylation > 50%). It is biocompatible because it can be in direct contact with living tissues and is biodegradable, since it is susceptible to attack from specific and non-specific enzymes, such as lysozyme, chitinase, cellulase or hemicellulase, protease, lipases, β-1,3-glucanase, and β-1,4-glucanase. In addition, it is considered as being non-toxic to humans or animals and is recognized as GRAS [55,56,57,58,59]. The functional properties of chitosan, a stable linear copolymer, depend on acetylated units (N-acetylglucosamine, and CH_3_CONH_2_, which form hydrogens and hydrophobic interactions) and amino groups (deacetylated glucosamine units, NH_3_, which in acidic media, becomes a polycationic molecule, which is an unusual property in a biopolymer). These characteristics allow it to interact with organic and inorganic molecules. The antifungal effect is mainly dependent on which chitosan can absorb proteins from the cell wall by weak electrostatic interactions. With phospholipids, the electrostatic interaction of the amino groups of chitosan and the phosphate groups of the phospholipids of the fungal membrane forms strong hydrophobic associations. With non-protonated amino groups, it has high affinity with most metals from cytoplasm, forming ionic and chelating interactions. These functional properties are based on the degree of acetylation and molecular weight, as well as purity, crystallinity, water content, and organic matter [60,61]. The antifungal effect of chitosan causes morphological alterations (deformation, corrugation, distortion, swelling and dehydration of hyphae, excessive branching, and increased vesicles), structural changes (shrinkage and disintegration of plasma membrane and extension of vacuoles) and cytoplasmic alterations (softening of cell wall, disorganization and disintegration of cytoplasm, and lysis and exit of cellular material) of the fungus [58,62,63].

Chitosan concentrations of 1.5–2.0% have been reported to completely inhibit *C. gloeosporioides* spore tube germination. This is due to them blocking the reception of external stimuli, which inhibits signal transduction to the nucleus. This in turn controls the gene expression of the formation of the germ tube of the spore [64,65]. The greatest effect of the deposition of chitosan is on spores rather than on mycelium [66]. Another example of this effect was that reported by Correa-Pacheco et al. [37], where application of chitosan nanoparticles in vitro reduced mycelial growth between 85 and 100%, reduced sporulation, and inhibited spore germination by 100%; in situ, they reduced the incidence of *C. gloeosporioides* without affecting fruit quality. Obianom et al. [37] applied chitosan at 1.5% on avocado fruits inoculated with *C. gloeosporioides*, reducing the incidence of anthracnose by more than 65%. They attributed this to chitosan and its antifungal and elicitation action due to the increase in the expression of genes that encode the PAL enzyme responsible for the production of salicylic acid, which in turn induces defense mechanisms. Kaleda-Marino et al. [67] applied combinations of chitosan and propolis in vitro and in vivo, inhibiting the growth of the fungus by more than 90%, as well as reducing the severity of anthracnose in fruits caused by *C. gloeosporioides*. They concluded that chitosan combined with propolis has an antifungal effect and maintains the quality of the fruit.

#### 3.4.1. Deposition of Chitosan

Chitosan deposition is based on the formation of dense films on the surface of the conidia or spores, limiting the metabolic processes and interaction with the medium. Cells with chitosan deposition appear to have a thick outer membrane. However, this film prevents the receptors of the fungal cell wall from perceiving the presence of nutrients, a change in pH, or the excretion of metabolites. Deposition is a mode of action of chitosan that is as effective as the antifungal effect [58,68,69].

#### 3.4.2. Chitosan as an Elicitor

The inducing effect of films and coatings based on chitosan in “Hass” avocado fruit is due to the fact that it is recognized by the PRRs (Pattern Recognition Receptors) in the cell wall as a biotic stress (MAMPs/PAMPs) or abiotic (DAMPs). After this recognition, ions diffuse rapidly through the plasma membrane, beginning the response through PR with the production of salicylic acid (green fruits—bitrophism) via phenylpropanoids in the cytosol. Phenylalanine is produced in chloroplasts and in the cytosol by the activity of PAL, and it is converted into trans-cinnamic acid, which becomes salicylic acid. In mature fruit (necrotrophism), it occurs through the production of jasmonic acid, a derivative of linoleic acid (C18, 18:3) released from the cell membrane, which is oxidized by LOX (lipoxygenase), which after cyclization, reduction, and β oxidation produces jasmonic acid. Both compounds activate the systemic acquired resistance (SAR) response hours or days after Plant Immunity Triggered (PIT) caused by chitosan [70,71,72,73]. In “Hass” avocado, the expression of genes of a large number of metabolic response processes regulated by the application of chitosan has been documented, which prevents the dispersal of *Colletotrichum* sp. The application of chitosan also achieved an inducing effect on genes that activate PAL, CHI, and an increase in the SOD enzyme in the control of anthracnose (*C. gloeosporioides*) and stem and rot (*L. theobromae*). The capacity and efficiency of chitosan to elicit these defense mechanisms are directly related to its physicochemical properties, molecular weight, degree of acetylation, and viscosity [58,62] (Table 1).

#### 3.4.3. Chitosan–Nucleus Interaction

Depending on the concentration and size of the chitosan, it can penetrate further and reach the nucleus, where it destabilizes the nuclear membrane and interacts with DNA and RNA, interfering with the vital functions of the fungus [62,74,75,76,77,78,79]. Finally, the internal and external morphologies of the conidia and mycelium of *C. gloeosporioides*, as well as the biochemical and physiological processes of the fungus, are altered without the possibility to sporulate [80].

**Table 1 molecules-26-06819-t001:** Main effects of GRAS application in vitro and on avocado fruit defense mechanism during storage.

	In Vitro		Fruit Defense Mechanism		Reference No.
GRAS Compound	Mycelial Growth Inhibition (%)	Antifungal Response	Defense Enzymes	Antioxidant Enzymes	
Essential oils	60–100	Production of phenols	CHI, 1,3-β-GLU, PAL, and POX	SOD and CAT	[36]
Essential oils		Production of monoterpene phenol derivative	Upregulation of PAL gene expression	Enhanced biosynthesis of epicatechin	[35]
*Bacillus* sp.	30–55	Production of volatile compounds			[47]
Volatile compounds	100		PAL, CHI, and β-1,3 GLU	Total phenolic contents	[52]
Chitosan			Upregulation of PAL and downregulation of LOX genes. Upregulation of CHI genes	Higher epicatechin contents and higher SOD activity	[67]
Chitosan			Induced unigenes related to systemic acquired resistance	Induction of genes involved in response to both biotic and abiotic stress	[76]
Silicon			CHI, 1,3-β-GLU, PAL, and POX	SOD and CAT	[63]

CHI = chitinase, 1,3-β-GLU = glucanase, PAL = phenylalanine ammonia-lyase, POX = peroxidase, LOX = lipoxygenase, SOD = superoxide dismutase, CAT = catalase.

**Table 2 molecules-26-06819-t002:** Main effects of GRAS application on avocado postharvest quality, disease, and pathogens involved during storage.

GRAS Compound	Postharvest Quality	Disease	Microorganism Involved	Reference No.
Essential oils		Anthracnose	*C. gloeosporioides*	[35,36]
*Bacillus* sp.		Fusarium dieback, anthracnosis, and Phytophthora root rot	*Fusarium solani*, *Fusarium* sp., *C. gloeosporioides*	[47]
Volatile compounds		Stem-end rot	*Lasiodiplodia theobromae*	[52]
Chitosan		Stem-end rot and anthracnose	*Lasiodiplodia theobromae*, *C. gloeosporioides*	[67]
Chitosan	Decreased respiratory rate, ethylene production, and fresh mass loss. Increased pulp firmness	Anthracnose	*C. gloeosporioides*	[65,76]
Chitosan	Reduced severity and incidence of anthracnose, maintained fruit quality	Anthracnose	*C. gloeosporioides*	[67]
Silicon	Decreased respiratory rate, ethylene production, and fresh mass loss	Anthracnose	*C. gloeosporioides*	[63]

### 3.5. Silicon

Silicon is one of the most abundant elements in the world and is presented as calcium silicate, potassium silicate, sodium metasilicate, and biosilicate. It is highly soluble in water, is low cost, and is classified as a GRAS substance due to its low toxicity with regard to humans, animals, and the environment [81,82]. The mode of action of silicon is based on forming a layer under the cuticle. This is due to the attraction of silicon to the organosilicon compounds that are firmly attached to the cell wall, forming an amorphous layer of silicate of 2.5 µm immediately under the cuticle (0.1 µm) on avocado exocarp fruit [83,84]. This layer inhibits the penetration of *C. gloeosporioides* and makes the cell less susceptible to the degradative enzymes of the cell wall excreted by the pathogen. Moreover, the haustorium is invaded by the silicate and prevents the production of infection hypha. Since silicon is bound to the hemicellulose of the cell wall, it is reinforced and regenerated. After forming the layer, the silicate binds to the hydroxyl groups of proteins related to the cell signaling cascade, activating defense mechanisms, such as the activity of defense enzymes, including CHI, 1, 3-β-GLU, PAL, POX, LOX, SOD, and CAT, on postharvest avocado fruit [85,86,87]. The postharvest application of silicon on avocado has been performed by immersion and at concentrations ranging from 100 to 25,000 ppm. Low concentrations have an elicitor/regulating effect on genes that activate enzymes related to defense mechanisms and increase the concentration of polyphenols against *C. gloeosporioides* [83]. Applications with potassium silicate between 2940 and 1470 ppm cross the exocarp to the mesocarp, forming a layer that decreases the respiration rate (CO_2_) due to gas exchange. In addition, the accumulation of antioxidants and total phenols in the “Hass” avocado fruit maintains their quality for long periods of refrigeration. However, preharvest applications have no effect on quality [88] (Table 2).

## 4. Future Perspectives

The commercial application of biostimulants is increasing exponentially. For instance, in Europe, over 6.2 million ha of land is treated per annum with these natural-derived molecules to increase yield and food quality [89]. However, their application is still limited due to inconsistent efficacy compared to synthetic agrochemicals, mostly due to the lack of knowledge with regard to the molecular mechanisms associated with the plant metabolism induced by their application [90]. Similarly, although the antifungal and elicitor effects of GRAS compounds is evident, it is necessary to identify and characterize the molecular mechanisms that are differentially induced when applied. Recently, the avocado genome has been sequenced [91,92]. This information opens the possibility of investigating, from a genome-wide point of view, the responses induced by GRAS molecules with regard to this important fruit. This could help us not only to understand the molecular mechanisms that activate the induced defense responses but to design and improve alternative processes with regard to the chemical control of pathogens.

## 5. Conclusions

This review shows the advances in the use of non-chemical treatments (natural or biological compounds) that have an antifungal effect and induce defense mechanisms of avocado fruit. Essential oils, *Bacillus* sp., volatile organic compounds, chitosan and silicon have been shown to possess these properties. In vitro, they reduce the viability of the fungus, and in fruits they regulate genes for the expression of response enzymes and antioxidants. In addition, they have been shown to extend the postharvest life of the fruit.

## Figures and Tables

**Figure 1 molecules-26-06819-f001:**
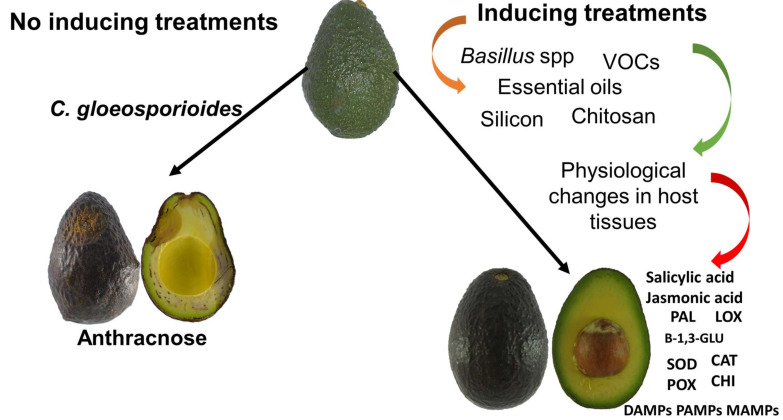
Anthracnose infection processes in preharvest, harvest, and postharvest until eating maturity without inducing treatments, and mechanisms of elicitation action of natural compounds in the fruit after their application to the fruit at harvest until eating maturity with inducing treatments.

## Data Availability

The data presented in this study are available on request from the corresponding author.

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
