# Peer review of "Non-Chemical Treatments for the Pre- and Post-Harvest Elicitation of Defense Mechanisms in the Fungi–Avocado Pathosystem"

_molecules, 2021, doi:10.3390/molecules26226819_

Round 1
Reviewer 1 Report
Please check the review comments in the pdf file

Author Response
Response to Reviewer 1 Comments
Point 1: Title: The title is talking about natural compounds for the postharvest elicitation of defense mechanisms . However, the manuscript discussed a few preharvest treatments rather than postharvest elicitation defense mechanisms. A more precise title would be better for this manuscript..
Response 1: the title was modified to the reviewer's observations for Point 1.
Point 2: Abstract
line22 23; line25 line22 23 has grammar issue, and line 25 is it an avocado fruit cell?
Line85 88; 128 135; 204 210; 250 254 290 293 and 301 305: these sections all have grammar issues and difficul t to read.
Response 2: a review of the grammatical structure was carried out for Point 2.
Point 3: Line 91: Please check the proper writing for single oxygen
Response 3: The correct form is 1O2 for Point 3.
Point 4: Line 146 169: Is Bacillus spp. a natural compound? Also, this whole section discusses about the preharvest treatment rather than postharvest elicitation. I t is not quite fit for the title of this manuscript.
Response 4: the title was modified to the reviewer's observations for Point 4.
Point 5: Page10 line 323 329: The conclusion is quite confusing. I t is and needs to be checked and rewrite.
Response 5: The conclusions were reviewed and rewritten for Point 5.
"Please see the attachment."

Reviewer 2 Report
The manuscript refers to the natural defense mechanisms that can be sustained by natural compounds in avocado, for the postharvest period against the fungal attack of Colletotrichum gloeosporioides.
The paper brings in the front the molecular mechanisms sustain by elicitors to modify the enzymatic status and triggering of the process of local or systemic resistance of fruits unlike other studies that refer to the physiological and qualitative aspects of avocados during storage.
The paper is well structured and elaborated with scientific language, in accordance with the latest research in the field. The bibliographic references, in number of 92, most of them are published in the last 6 years.
I agree with publication.
Suggestion for authors
There are inconsistencies in the bibliographic citations in the text and the references at the end of the manuscript (see reference 92).
Author Response
Response to Reviewer 2 Comments
Point 1: Suggestion for authors
There are inconsistencies in the bibliographic citations in the text and the references at the end of the manuscript (see reference 92).
Response 1: The bibliography was revised and modified for Point 1.
Please see the attachment.

Round 2
Reviewer 1 Report
I have no further comments.